# Relationships among Depressive Symptoms, Body Weight, and Chronic Pain: A Cross-Sectional Analysis of the Shika Study

**DOI:** 10.3390/bs13020086

**Published:** 2023-01-20

**Authors:** Shinobu Fukushima, Fumihiko Suzuki, Hiromasa Tsujiguchi, Akinori Hara, Sakae Miyagi, Takayuki Kannon, Keita Suzuki, Yukari Shimizu, Thao Thi Thu Nguyen, Toru Yanagisawa, Fumika Oku, Kuniko Sato, Masaharu Nakamura, Koichiro Hayashi, Aki Shibata, Tadashi Konoshita, Yasuhiro Kambayashi, Hirohito Tsuboi, Atsushi Tajima, Hiroyuki Nakamura

**Affiliations:** 1Department of Public Health, Graduate School of Advanced Preventive Medical Sciences, Kanazawa University, 13-1 Takaramachi, Kanazawa 920-8640, Japan; 2Department of Hygiene and Public Health, Faculty of Medicine, Institute of Medical, Pharmaceutical and Health Sciences, Kanazawa University, Kanazawa 920-8640, Japan; 3Community Medicine Support Dentistry, Ohu University Hospital, Koriyama 963-8611, Japan; 4Advanced Preventive Medical Sciences Research Center, Kanazawa University, 1-13 Takaramachi, Kanazawa 920-8640, Japan; 5Innovative Clinical Research Center, Kanazawa University, 13-1 Takaramachi, Kanazawa 920-8641, Japan; 6Department of Bioinformatics and Genomics, Graduate School of Advanced Preventive Medical Sciences, Kanazawa University, 13-1 Takaramachi, Kanazawa 920-8640, Japan; 7Department of Nursing, Faculty of Health Sciences, Komatsu University, 14-1 Mukaimotorimachi, Komatsu 923-0961, Japan; 8Faculty of Public Health, Hai Phong University of Medicine and Pharmacy, Ngo Quyen, Hai Phong 180000, Vietnam; 9Department of Clinical Cognitive Neuroscience, Graduate School of Medical Science, Kanazawa University, Kakuma-machi, Kanazawa 920-1192, Japan; 10Third Department of Internal Medicine, Faculty of Medical Sciences, University of Fukui, Shimoaizuki, Eiheiji-cho, Yoshida-gun, 23-3 Matsuoka, Fukui 910-1193, Japan; 11Department of Public Health, Faculty of Veterinary Medicine, Okayama University of Science, 1-3 Ikoinooka, Imabari 794-8555, Japan; 12Graduate School of Human Nursing, University of Shiga Prefecture, 2500 Hassaka-cho, Hikone 522-8533, Japan

**Keywords:** chronic pain, body mass index, geriatric depression scale, cross-sectional study, the Breslow–Day test

## Abstract

Although depression and body weight have individually been associated with chronic pain (CP), it currently remains unclear whether the combination of depressive symptoms (DS) and being underweight/overweight is related to CP. Therefore, we herein investigated the relationships among depression, body mass index (BMI), and CP in community-dwelling middle-aged and elderly individuals. Participants comprised 2216 inhabitants of Shika town in Ishikawa prefecture, Japan, including 1003 males (mean age of 68.72 years, standard deviation (SD) of 8.36) and 1213 females (mean age of 69.65 years, SD of 9.36). CP and DS were assessed using a CP questionnaire and Geriatric Depression Scale-15, respectively. The Breslow–Day test indicated that DS positively correlated with lumbar/knee pain in the BMI < 25 group, but not in the BMI ≥ 25 group. Furthermore, lumber/knee pain was related to a higher BMI. These results were confirmed by a logistic analysis with age, sex, BMI, solitary living, the duration of education, no exercise/hobbies, smoking history, alcohol intake, and medical treatment for diabetes, hyperlipidemia, or hypertension as confounding factors. The present study indicates the importance of considering DS and BMI in the prevention of CP. Further studies are needed to clarify the causal relationships among depression, BMI, and CP.

## 1. Introduction

Pain is defined by the International Association for the Study of Pain (IASP) as an unpleasant sensory and emotional experience associated with, or resembling that associated with, actual or potential tissue damage [1]. Chronic pain (CP) is defined as pain that persists or recurs for more than three months [2]. The prevalence of CP ranges between 17.5% [3] and 40.2% [4] in Japan, and 50.2 million (20.5%) adults in the USA have CP [5]. Primary prevention that targets community residents is important because CP affects the quality of life (QOL) of adults [6].

Our previous studies reported relationships between CP and various factors, such as vitamin intake and health-related QOL (HRQoL) [7], hypertension and HRQoL [8], alcohol intake and depression [9], fatty acid intake and C-reactive protein levels [10], and hypertension and being underweight [11]. Stress caused by pain has been implicated in the development of depression [12]. A clinical study showed that up to 85% of CP patients had severe depression [13], while a cross-sectional study reported that the threshold for CP was significantly reduced in patients with depression [14]. A common mechanism in depressive disorder and CP is central hyperexcitability caused by inhibited serotonergic functions [14]. Therefore, the relationship between CP and depression warrants further study.

In addition to CP, depression is associated with body weight [15,16,17,18,19,20,21,22,23]. A cross-sectional study reported a positive relationship between being overweight or obese and depression and depression-like symptoms in both sexes [22]. In contrast, another study demonstrated that individuals with depression-like symptoms were likely to be not only obese, but also underweight [21].

Another factor related to CP is body weight. A high body mass index (BMI) was previously shown to correlate with chronic low back pain [24,25,26]. However, limited information is currently available on the relationships among depression, body weight, and CP [27]. A cross-sectional study revealed that age, BMI, and depression positively correlated with the risk and severity of CP [28]. Furthermore, a co-twin study by Pradeep et al. [18] and a cross-sectional study by Rogers et al. [29] indicated relationships among obesity, depression, and CP. Zheng et al. [30] recently suggested that a younger age, higher BMI, higher scores for Western Ontario and McMaster Universities Osteoarthritis Index (WOMAC) pain, dysfunction, and stiffness, a lower education level, the presence of more than one comorbidity, and two or more body sites with pain correlated with a higher incidence of depression.

Although depression and body weight have individually been associated with CP, it currently remains unclear whether the combination of depressive symptoms (DS) and being underweight/overweight are related to CP. Therefore, we herein investigated the relationships among depression, BMI, and CP in community-dwelling middle-aged and elderly individuals.

## 2. Materials and Methods

### 2.1. Participants

We leveraged cross-sectional data collected from October 2013 to December 2016 in the Shika Study [7,8,9,10,31,32,33,34]. The subjects were 5013 residents [35] aged 40 years or older living in four randomly selected model districts (Horimatsu, Higashimasuho, Tsuchida, and Togi districts) in Shika town, Ishikawa prefecture in Japan, and obtained informed consent from all. Of the 4550 people who underwent physical examination, 2116 subjects (1003 males and 1213 females with mean ages of 68.72 years (SD = 8.36) and 69.65 years (SD = 9.36), respectively) were selected for analysis according to the following selection criteria. Selection criteria: answered questions related to age, education, BMI, living pattern, exercise/hobbies, smoking, alcohol consumption, and the treatment of underlying diseases; age at the time of these answers was 55 years or older; energy intake of more than 600 kcal/day and less than 4000 kcal/day; and answered the CP questionnaire and Geriatric Depression Scale-15 (GDS-15). Figure 1 shows the participant recruitment chart.

### 2.2. Instrumentation

#### 2.2.1. CP

CP was evaluated using the same method as in the study by Amatsu et al. [9]. That is, participants were asked whether they had pain that lasted more than 3 months, its location, and the intensity of that pain on a scale of 1–10.

#### 2.2.2. DS Assessment

The Japanese short version of GDS-15 for self-completed questionnaire [36,37] was used to assess depressive symptom states, and evaluated as in the study by Amatsu et al. [9].

#### 2.2.3. BMI

BMI was calculated using current weight (kg) and height (meters).

### 2.3. Questionnaire on Characteristics

Participants completed a self-administered questionnaire on age, sex, the duration of education, exercise/hobbies, smoking history, and medical treatment for diabetes, hyperlipidemia, hypertension, and depression.

### 2.4. Statistical Analysis

Statistical analyses were conducted with IBM SPSS Statistics ver. 25 for Windows (IBM, Armonk, NY, USA). The Student’s *t*-test was applied to investigate the relationships between continuous variables; on the other side, the chi-square test was adopted to examine those between categorical variables. The Breslow–Day test was used to analyze the homogeneity of proportions of CP for each site by BMI and DS. Multiple logistic regression analysis was conducted for relationship assessment between DS and pain at any site, neck/shoulder/upper limb pain, and lumbar/knee pain stratified by BMI. Independent variables were age, sex, BMI, solitary living, the duration of education, no exercise/hobbies, smoking history, alcohol intake (density method), and medical treatment for diabetes, hyperlipidemia, or hypertension. The data were stratified by BMI and analyzed. Forced entry method was used to select variables. *p* < 0.05 was considered statistically significant.

## 3. Results

### 3.1. Participant Characteristics

Table 1 indicates the characteristics of participants by sex, including BMI, DS, and CP (Table 1). Females were significantly older than males (*p* < 0.001). The percentages of participants with solitary living (*p* < 0.001), without exercises/hobbies (*p* = 0.026), and undergoing treatment for hyperlipidemia (*p* < 0.001) were significantly lower in males than in females. In contrast, the duration of education (*p* = 0.006), the percentage of smokers (*p* < 0.001), and those undergoing treatment for diabetes (*p* < 0.001) and hypertension (*p* = 0.006) were significantly lower in females than in males. Alcohol intake (crude data and density method) was also significantly lower in females than in males (*p* < 0.001). On the other hand, the percentage of participants with no alcohol intake was lower in males than in females (*p* < 0.001). The percentages of participants with pain at any site (*p* = 0.005) and lumber/knee pain (*p* = 0.043) were significantly higher in females than in males. No significant differences were observed in the percentages of participants with DS (*p* = 0.064) and BMI (*p* = 0.272) between males and females.

### 3.2. Comparisons of Participants without and with DS

Table 2 presents comparisons of participants without and with DS. The mean age was significantly younger in the no DS group than in the DS group (*p* < 0.001). The percentages of participants with solitary living (*p* = 0.0289) and exercise/hobbies (without) (*p* < 0.001) were significantly lower in the no DS group than in the DS group. The duration of education (*p* = 0.013), BMI (*p* = 0.001), the percentage of participants with treatment for hyperlipidemia (*p* < 0.001), and the percentage of participants without alcohol intake (*p* = 0.025) were significantly lower in the DS group than in the no DS group. The percentages of participants with pain at any site (*p* < 0.001), neck/shoulder/upper limb pain (*p* = 0.016), and lumber/knee pain (*p* < 0.001) were significantly lower in the no DS group than in the DS group.

### 3.3. Comparisons of Participants with and without Lumber/Knee Pain

Table 3 shows comparisons of participants with and without lumber/knee pain. Mean age was significantly older in the no lumber/knee pain group than in the lumber/knee pain group (*p* < 0.001). The percentage of participants with solitary living (*p* = 0.003), DS (*p* < 0.001), no alcohol intake (*p* = 0.011), and BMI (*p* = 0.047) were significantly lower in the no lumber/knee pain group than in the lumber/knee pain group. The duration of education was significantly shorter in the lumber/knee pain group than in the no lumber/knee pain group (*p* < 0.001). The percentages of participants with pain at any site (*p* < 0.001), neck/shoulder/upper limb pain (*p* < 0.001), and foot pain (*p* < 0.001) were significantly lower in the no lumber/knee pain group than in the lumber/knee pain group.

### 3.4. Comparisons of CP and DS in Participants Stratified by two BMI Groups

Table 4 shows the percentage of participants with pain at any site, neck/shoulder/upper limb pain, lumber/knee pain, and foot pain with and without DS stratified by BMI < 25 and ≥ 25. Chi-square tests showed that the percentages of participants with pain at any site (*p* < 0.001), neck/shoulder/upper limb pain (*p* = 0.016), lumber/knee pain (*p* < 0.001), and foot pain (*p* = 0.010) were significantly lower in the DS with BMI < 25 group than in the no DS with BMI < 25 group, whereas no significant differences were observed between the DS and no DS with BMI ≥ 25 groups. With Breslow–Day tests, an analysis of homogeneity for the percentage of participants having lumber/knee pain with DS between the BMI groups showed a significant result (*p* = 0.049). Therefore, the percentage of participants having lumber/knee pain with DS was significantly higher in BMI < 25 group than in the BMI ≥ 25 group.

### 3.5. Multiple Logistic Regression Analysis of DS and CP in Participants Stratified by Two BMI Groups

Table 5 shows the results of a multiple logistic regression analysis of the relationship between CP and DS in participants stratified by BMI. The independent variable was DS adjusted for the following covariates: age, sex, BMI, solitary living, the duration of education, exercise/hobbies (without), smoking, alcohol intake (density method), DS, and treatment for diabetes, hyperlipidemia, and hypertension. DS were a significant independent variable for pain at any site (OR: 2.191; 95%CI: 1.526, 3.144; *p* < 0.001), neck/shoulder/upper limb pain (OR: 2.040; 95%CI: 1.156, 3.600; *p* = 0.014), and lumber/knee pain (OR: 2.195; 95%CI: 1.437, 3.353; *p* < 0.001) in participants with BMI < 25, but not in those with BMI ≥ 25. Furthermore, age was a significant independent variable for pain at any site (OR: 1.030; 95%CI: 1.008, 1.052; *p* = 0.007), neck/shoulder/upper limb pain (OR: 0.952; 95%CI: 0.918, 0.987; *p* = 0.008), and lumber/knee pain (OR: 1.059; 95%CI: 1.033, 1.086; *p* < 0.001), sex was a significant independent variable for pain at any site (OR: 2.231; 95%CI: 1.411, 3.527; *p* = 0.001), neck/shoulder/upper limb pain (OR: 2.180; 95%CI:1.030, 4.612; *p* = 0.042), and lumber/knee pain (OR: 1.766; 95%CI: 1.054, 2.957; *p* = 0.031), and BMI was a significant independent variable for lumber/knee pain (OR: 1.122; 95%CI: 1.013, 1.243; *p* = 0.027) in participants with BMI < 25, but not in those with BMI ≥ 25. The duration of education was a significant variable for pain at any site (OR: 0.874; 95%CI: 0.771, 0.992; *p* = 0.037) and smoking was a significant variable for pain at any site (OR: 2.736; 95%CI: 1.258, 5.952; *p* = 0.011) and lumber knee pain (OR: 4.950; 95%CI: 2.102, 11.656; *p* < 0.001) in participants with BMI ≥25, but not in those with BMI < 25. These results suggest that CP is associated with DS only in participants with BMI < 25.

## 4. Discussion

The main result of the present study was that DS positively correlated with lumbar/knee pain in the BMI < 25 group, but not in the BMI ≥ 25 group. Furthermore, lumber/knee pain was related to a higher BMI. Previous studies indicated that individuals with a high BMI are more likely to have CP. A cross-sectional study by Lee et al. showed that weight gain correlated with low back pain [24]. A meta-analysis by Shiri et al. [25] revealed that being overweight or obese correlated with chronic low back pain. In addition, a Mendelian randomized trial by Elgaeva et al. [26] demonstrated the causal effect of BMI on chronic low back pain. A cross-sectional study with a multivariate model by Dong et al. [38] indicated that a BMI of 30 or higher was a modifiable factor in older adults, whereas a BMI lower than 30 was not. The present results showed that BMI was significantly higher in the lumbar/knee pain group than in the non-lumbar/knee pain group. Therefore, BMI appears to be a risk factor for CP. On the other hand, a two-way ANCOVA of the interaction between BMI groups and DS groups for CP revealed that CP was more frequent among participants in the BMI < 25 group with DS. The discrepancy between the findings of Lee et al. [24] and the present results may be attributed to the inclusion of DS in the analysis; we examined the involvement of not only BMI, but also DS in CP. We considered a high BMI to be directly associated with chronic low back pain; however, it is important to note that even a low BMI was associated with chronic low back pain in the presence of DS.

A review by Sheng et al. on the relationship between CP and depression reported that 85% of CP patients had severe depression [13]. A cross-sectional study by Klausenberg et al. [14] demonstrated that the threshold for CP was significantly reduced in patients with depression. The present study also showed that participants with DS were more likely to have CP, but that this relationship was linked to BMI. Regarding the relationship between depression and BMI, a longitudinal study by Kim et al. [39] indicated that depression contributed to weight loss rather than obesity, and also that being underweight appeared to result in the development of DS in middle-aged and elderly Asian populations. On the other hand, a cross-sectional study by Badillo et al. [20] suggested a positive relationship between being overweight or obese and depression and depression-like symptoms in both sexes. In the present study, BMI was lower in the DS group than in no DS group. Therefore, in consideration of the different results obtained on the relationship between CP and DS and CP and BMI, further studies are needed to concurrently examine the relationships among CP, DS, and BMI. The relationships among CP, depression, and BMI have only been examined in a cross-sectional analysis by Häuser et al. [27], and the findings obtained showed that age, BMI, and depression independently predicted disabling chronic low back pain. The present results demonstrated that DS correlated with chronic lumber/knee pain in the BMI < 25 group, but not in the BMI ≥ 25 group. Therefore, we speculate that DS are related to CP in non-obese individuals.

As a mechanism for low back pain, obesity with a BMI ≥ 25 is speculated to be related to weight compression of the discs in the joints [40] and anterior abdominal curvature of the lumbar spine [41]. In addition, cytokines secreted from adipose tissue and an increased adipocyte size, which is a characteristic of obesity, cause pain-mediated changes in the neurophysiological properties of peripheral nociceptors and central neurons [24,42]. On the other hand, in subjects with BMI < 25, central hyperexcitation due to decreased serotonin-producing function is involved in pain [14], and reduced serotonin/noradrenaline activates descending circuits at the spinal level to reduce the pain threshold [43], suggesting that blocked descending inhibitory pathways are associated with CP, even in subjects who are not overweight. Therefore, the mechanisms of CP development are different for obesity and DS. On the other hand, our results showed that DS is a factor involved in lumbar/knee pain in non-obese subjects, whereas the presence or absence of DS does not make a difference in the prevalence of lumbar/knee pain in obese subjects. Our findings showed that although obesity and DS are associated with CP, the presence of both factors does not additively or synergistically increase their prevalence. It is considered unique to this study since it has not been shown in previous studies. However, the involvement of these mechanisms requires further detailed investigation.

The present results showed that sex, the duration of education, and smoking were also associated with CP and BMI. Although a cross-sectional study by Hattori [44] demonstrated that sex was not associated with CP, the results obtained herein revealed that in the BMI < 25 group, women were more likely to have pain at any site, neck/shoulder/upper limb, and lumbar/knee pain than men. This difference may be related to the lower mean age in Hattori’s study [45] than in the present study and a different mechanism for the development of CP. Regarding the relationship between the duration of education and CP, a cross-sectional study reported that a lower level of education correlated with a higher prevalence of chronic low back and knee pain [44]. The present results also showed that the duration of education was lower in the CP group than in the non-CP group. Regarding smokers and CP, smokers report more severe pain and functional impairment than non-smokers [46,47,48]. The cessation of smoking and weight loss are used in combination to treat CP [49]. The present results were consistent with previous findings because CP was more likely to occur in participants with BMI ≥ 25 and a smoking history, but not in those with BMI < 25. These differences were significant for lumbar/knee pain. Our results show that solitary living is relevant in both CP and DS. Previous studies have reported an association between CP and loneliness [50,51,52]. A cross-sectional study of older adults [50] and a prospective longitudinal study of middle-aged [51] have demonstrated the correlation between pain and loneliness with depression. Furthermore, a longitudinal study by Emerson et al. [52] reported that at both the onset of the study and four years later, those in pain had 1.58 times higher levels of loneliness than those who were not. Therefore, our results seem to support the findings of previous studies.

### Limitations and Future Directions

The strength of the present study is that it is an almost all-inclusive survey. On the other hand, it was a cross-sectional analysis; therefore, causal relationships regarding DS, BMI, and CP cannot be elucidated. In addition, no objective measurements were conducted for BDHQ, GDS, and CP because they are all self-administered surveys.

## 5. Conclusions

The present study indicated that DS positively correlated with lumbar/knee pain in the BMI < 25 group, but not in the BMI ≥ 25 group. Furthermore, lumber/knee pain was related to a higher BMI. The present study indicates the importance of considering DS and BMI in the prevention of CP.

## Figures and Tables

**Figure 1 behavsci-13-00086-f001:**
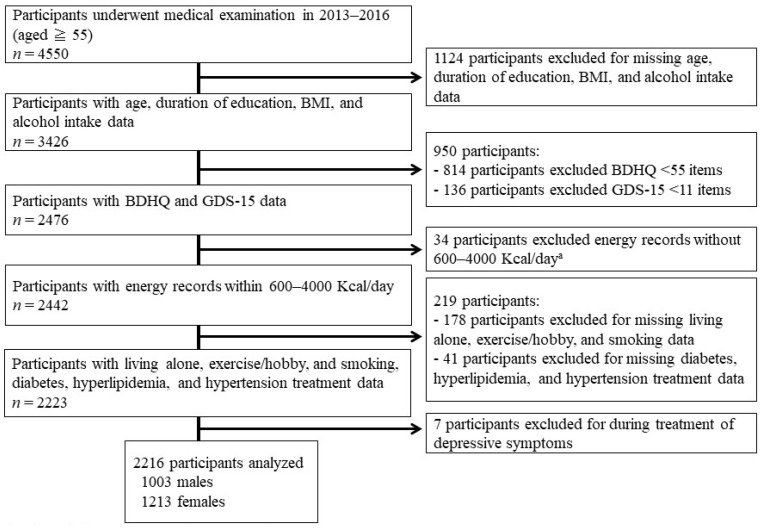
Participant recruitment chart. Abbreviations: BMI, Body Mass Index; BDHQ, Brief-type self-administered Diet History Questionnaire; GDS, Geriatric Depression Scale.

**Table 1 behavsci-13-00086-t001:** Participant characteristics.

	Male (*n* = 1003)		Female (*n* = 1213)		*p*-Value ^a^
Age, years, mean (SD)	68.72	8.36	69.65	9.36	<0.001
Solitary living, *n* (%)	75	7.48	150	12.37	<0.001
Education, years, mean (SD)	11.40	3.02	10.88	2.50	0.006
Exercise/hobbies (without), *n* (%)	597	59.52	778	64.14	0.026
Smoking history, *n* (%)	283	28.22	43	3.54	<0.001
BMI, kg/m^2^, mean (SD)	23.40	2.96	22.64	3.23	0.064
Treatment for diabetes, *n* (%)	131	13.06	93	7.67	<0.001
Treatment for hyperlipidemia, *n* (%)	110	10.97	217	17.89	<0.001
Treatment for hypertension, *n* (%)	362	36.09	371	30.59	0.006
DS, *n* (%)	348	34.70	394	32.48	0.272
Alcohol (crude data), g, mean (SD)	19.92	25.11	2.01	6.29	<0.001
Alcohol (density method), % energy, mean (SD)	3.81	4.54	0.503	1.54	<0.001
No alcohol intake, *n* (%)	322	32.10	906	74.69	<0.001
Alcohol intake, *n* (%)	681	67.90	307	25.31	
CP					
Any, *n* (%)	73	7.28	130	10.72	0.005
Head, *n* (%)	0	-	4	0.330	0.090
Neck/shoulder/upper limb, *n* (%)	27	2.69	46	3.79	0.149
Lumbar/knee, *n* (%)	56	5.58	94	7.75	0.043
Foot, *n* (%)	15	1.50	23	1.90	0.470

^a^ For *p*-values, Student’s *t*-test was used for mean ± standard deviation, and the chi-square test was used for *n* numbers and percentages. Abbreviations: SD, standard deviation; BMI, body mass index; DS, depressive symptoms; CP, chronic pain.

**Table 2 behavsci-13-00086-t002:** Characteristics of hypertension and GDS according to sex.

	Without DS(*n* = 1474)		With DS(*n* = 742)		*p*-Value ^a^
Sex (female), *n* (%)	819	55.56	394	53.10	0.272
Age, years, mean (SD)	68.68	8.18	70.32	10.18	<0.001
Solitary living, *n* (%)	135	9.16	90	12.13	0.029
Education, years, mean (SD)	11.22	2.84	10.91	2.58	0.012
Exercise/hobbies (without), *n* (%)	821	55.70	554	74.66	<0.001
Smoking history, *n* (%)	213	14.45	113	15.23	0.625
BMI, kg/ m^2^, mean (SD)	23.11	3.00	22.74	3.37	0.001
Treatment for diabetes, *n* (%)	147	9.97	77	10.38	0.766
Treatment for hyperlipidemia, *n* (%)	249	16.89	78	10.51	<0.001
Treatment for hypertension, *n* (%)	500	33.92	233	31.40	0.234
Alcohol (crude data), g, mean (SD)	10.43	19.56	9.49	19.85	0.288
Alcohol (density method), % energy, mean (SD)	2.04	3.57	1.93	3.81	0.492
No alcohol intake, *n* (%)	792	53.73	436	58.76	0.025
Alcohol intake, *n* (%)	682	46.27	306	41.24	
CP					
Any, *n* (%)	104	7.06	99	13.34	<0.001
Head, *n* (%)	1	0.07	3	0.40	0.112
Neck/shoulder/upper limb, *n* (%)	39	2.65	34	4.58	0.016
Lumbar/knee, *n* (%)	76	5.16	74	9.97	<0.001
Foot, *n* (%)	19	1.29	19	2.56	0.030

^a^ For *p*-values, Student’s *t*-test was used for mean ± standard deviation, and the chi-square test was used for *n* numbers and percentages. Abbreviations: DS, depressive symptoms; SD, standard deviation; BMI, body mass index; CP, chronic pain.

**Table 3 behavsci-13-00086-t003:** Characteristics of body weight and GDS according to sex.

	Non-CP (Lumbar/Knee) (*n* = 2066)		CP (Lumbar/Knee) (*n* = 150)		*p*-Value ^a^
Sex (female), *n* (%)	1119	54.16	94	62.67	0.043
Age, years, mean (SD)	68.88	8.75	73.99	10.00	<0.001
Solitary living, *n* (%)	199	9.63	26	17.33	0.003
Education, years, mean (SD)	11.19	2.77	10.11	2.36	<0.001
Exercise/hobbies (without), *n* (%)	1272	61.57	103	68.67	0.084
Smoking history, *n* (%)	299	14.47	27	18.00	0.239
BMI, kg/ m^2^, mean (SD)	22.95	3.12	23.48	3.33	0.047
Treatment for diabetes, *n* (%)	207	10.02	17	11.33	0.606
Treatment for hyperlipidemia, *n* (%)	310	15.00	17	11.33	0.221
Treatment for hypertension, *n* (%)	684	33.11	49	32.67	0.912
DS, *n* (%)	668	32.33	74	49.33	<0.001
Alcohol (crude data), g, mean (SD)	10.31	19.81	7.53	17.31	0.062
Alcohol (density method), % energy, mean (SD)	2.03	3.66	1.58	3.56	0.147
No alcohol intake, *n* (%)	1130	54.70	98	65.33	0.011
Alcohol intake, *n* (%)	936	45.30	52	34.67	
CP					
Any, *n* (%)	53	2.57	150	100.00	<0.001
Head, *n* (%)	3	0.15	1	0.67	0.245
Neck/shoulder/upper limb, *n* (%)	37	1.79	36	24.00	<0.001
Foot, *n* (%)	12	0.58	26	17.33	<0.001

^a^ For *p*-values, Student’s *t*-test was used for mean ± standard deviation, and the chi-square test was used for *n* numbers and percentages. Abbreviations: SD, standard deviation; BMI, body mass index; DS, depressive symptoms; CP, chronic pain.

**Table 4 behavsci-13-00086-t004:** Breslow–Day test on CP location and DS stratified by BMI.

			Non-CP	CP	Chi-Square Value	*P1*	*P2*
			*n* (%)	*n* (%)			
BMI < 25	Any	NDS	1048 (93.9%)	68 (6.1%)	25.806	<0.001	0.084
		DS	490 (86.6%)	76 (13.4%)			
	Neck/shoulder/ upper limb	NDS	1089 (97.6%)	27 (2.4%)	5.818	0.016	0.523
		DS	540 (95.4%)	26 (4.6%)			
	Lumber/knee	NDS	1069 (95.8%)	47 (4.2%)	22.225	<0.001	0.049
		DS	509 (89.9%)	57 (10.1%)			
	Foot	NDS	1106 (99.1%)	10 (0.9%)	6.644	0.010	0.193
		DS	552 (97.5%)	14 (2.5%)			
BMI ≥ 25	Any	NDS	322 (89.9%)	36 (10.1%)	1.089	0.297	0.084
		DS	153 (86.9%)	23 (13.1%)			
	Neck/shoulder/ Upper limb	NDS	346 (96.6%)	12 (3.4%)	0.466	0.495	0.523
		DS	168 (95.5%)	8 (4.5%)			
	Lumber/knee	NDS	329 (91.9%)	29 (8.1%)	0.364	0.546	0.049
		DS	159 (90.3%)	17 (9.7%)			
	Foot	NDS	349 (97.5%)	9 (2.5%)	0.049	0.824	0.193
		DS	171 (97.2%)	5 (2.8%)			

P1: Chi-square test, P2: Breslow–Day test. Abbreviations: CP, chronic pain; BMI, body mass index; NDS, non-depressive symptoms; DS, depressive symptoms.

**Table 5 behavsci-13-00086-t005:** Relationship between CP location and DS stratified by BMI.

		Any			Neck/Shoulder/Upper Limb			LUMBAR/KNEE		
		Exp (B)	95%CI (Lower, Upper)	*p*-Value	Exp (B)	95%CI (Lower, Upper)	*p*-Value	Exp (B)	95%CI (Lower, Upper)	*p*-Value ^a^
BMI < 25 (*n* = 1682)	Age	1.030	1.008, 1.052	0.007	0.952	0.918, 0.987	0.008	1.059	1.033, 1.086	<0.001
	Sex	2.231	1.411, 3.527	0.001	2.180	1.030, 4.612	0.042	1.766	1.054, 2.957	0.031
	BMI	1.056	0.969, 1.151	0.213	0.971	0.848, 1.112	0.671	1.122	1.013, 1.243	0.027
	Solitary living	1.330	0.809, 2.186	0.261	1.241	0.541, 2.849	0.610	1.289	0.725, 2.292	0.388
	Duration of education	0.929	0.857, 1.008	0.078	0.874	0.771, 0.991	0.035	0.951	0.865, 1.046	0.304
	Exercise/ hobbies (without)	1.066	0.722, 1.576	0.747	0.821	0.457, 1.478	0.511	1.157	0.724, 1.850	0.541
	Smoking history	1.388	0.786, 2.451	0.259	1.106	0.465, 2.630	0.820	1.607	0.831, 3.110	0.159
	Alcohol (density method)	1.041	0.983, 1.101	0.168	1.052	0.969, 1.142	0.224	0.997	0.925, 1.074	0.931
	DS	2.191	1.526, 3.144	<0.001	2.040	1.156, 3.600	0.014	2.195	1.437, 3.353	<0.001
	Treatment for diabetes	1.046	0.564, 1.942	0.886	0.493	0.117, 2.075	0.335	1.159	0.590, 2.279	0.668
	Treatment for hyperlipidemia	0.558	0.289, 1.079	0.083	0.763	0.288, 2.020	0.586	0.584	0.270, 1.262	0.171
	Treatment for hypertension	0.816	0.539, 1.235	0.336	0.776	0.382, 1.574	0.482	0.808	0.502, 1.302	0.381
BMI ≥ 25 (*n* = 534)	Age	1.024	0.985, 1.065	0.224	1.002	0.939, 1.070	0.951	1.040	0.996, 1.086	0.078
	Sex	1.382	0.696, 2.746	0.355	1.081	0.359, 3.258	0.890	1.707	0.761, 3.830	0.195
	BMI	1.007	0.886, 1.145	0.916	0.883	0.682, 1.144	0.347	1.049	0.913, 1.205	0.501
	Solitary living	1.615	0.716, 3.642	0.248	0.511	0.064, 4.050	0.525	2.173	0.932, 5.068	0.072
	Duration of education	0.874	0.771, 0.992	0.037	0.955	0.791, 1.153	0.634	0.893	0.778, 1.024	0.104
	Exercise/ hobbies (without)	0.974	0.528, 1.796	0.933	1.405	0.501, 3.943	0.518	0.761	0.386, 1.501	0.431
	Smoking history	2.736	1.258, 5.952	0.011	2.425	0.741, 7.940	0.143	4.950	2.102, 11.656	<0.001
	Alcohol (density method)	1.008	0.930, 1.093	0.845	0.991	0.863, 1.137	0.892	1.009	0.923, 1.104	0.844
	DS	1.253	0.697, 2.251	0.451	1.348	0.518, 3.506	0.541	1.145	0.586, 2.236	0.692
	Treatment for diabetes	1.260	0.583, 2.724	0.556	1.723	0.565, 5.253	0.339	0.898	0.347, 2.321	0.824
	Treatment for hyperlipidemia	0.923	0.436, 1.951	0.833	1.883	0.623, 5.690	0.262	1.050	0.456, 2.419	0.908
	Treatment for hypertension	1.034	0.575, 1.860	0.910	0.804	0.300, 2.159	0.665	1.062	0.549, 2.058	0.857

^a^ Multiple logistic regression analysis. Abbreviations: Exp (B), Exponentiation of the B coefficient; CI, confidence interval; CP, chronic pain; BMI, body mass index; DS, depressive symptoms.

## Data Availability

Data in the present study are available upon request from the corresponding author. Data are not publicly available due to privacy and ethical policies.

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
