# Peer review of "Relationships among Depressive Symptoms, Body Weight, and Chronic Pain: A Cross-Sectional Analysis of the Shika Study"

_behavsci, 2023, doi:10.3390/bs13020086_

Round 1

Reviewer 1 Report

The paper address an important issue such as the combination of depressive symptoms and being underweight/overweight is related to CP.

Moreover, it enrolled a huge number of patients.

The major concern I have is on the originality of your data. In the discussion You stated  " depressive symptoms positively correlated with lumbar/knee pain in the BMI <25 group, but not in the BMI ≥25 group." How this finding could influence chronic pain? How do you explain this finding? Moreover, since BMI correlates with chronic pain (as you stated) how these factors interplay each other?

In other word, in obese people the influence of weight on chronic pain can exceed the absence of depressive symptoms?

I think that this is the centre of the problem and perhaps the main result of your research, therefore, it should be better discussed and analysed. 

Other results you found, such as " lumber/knee pain was related to a higher BMI. These results were confirmed by a logistic analysis with age, sex,, the duration of education, are all associated with CP and BMI" or "The present study indicates the importance of considering depressive symptoms and BMI in the prevention of CP" are clearly very important and fully documented in your research, but they are clearly demonstrated in thousands of papers, so maybe you should focus on the originality of your results.

Tables  1-2-3  are difficult to interpret since mean sd, n and percentage are not clearly stated. Moreover  check your data (alcohol users and non users for example did not fit with the total sample). Aldo the total number of patients with any pain does not fit with the sum of each site of pain.

Please check.

Reviewer 2 Report

The reviewed paper entitled “Relationships among Depressive Symptoms, Body Weight, and Chronic Pain: A Cross-sectional Analysis of the Shika Study” presents interesting results of studies conducted in a group of elderly women (1213) and men (1003). From the point of view of the methodology, the study was performed correctly. The test procedure is presented in a clear and comprehensible manner, and the statistical methods are adequate to the purpose of the study and the structure of the group. The results are presented and discussed in a clear way. The authors included also solitary living as one of the variables in the logistic analysis. Therefore, it would be worth referring in the discussion to the publications concerning the correlations between loneliness and chronic pain and depression, e.g.:

·         Emerson K, Boggero I, Ostir G, Jayawardhana J. Pain as a Risk Factor for Loneliness Among Older Adults. J Aging Health. 2018 Oct;30(9):1450-1461. doi: 10.1177/0898264317721348. Epub 2017 Jul 20. PMID: 28728466.

·         Wilson JM, Colebaugh CA, Meints SM, Flowers KM, Edwards RR, Schreiber KL. Loneliness and Pain Catastrophizing Among Individuals with Chronic Pain: The Mediating Role of Depression. J Pain Res. 2022 Sep 16;15:2939-2948. doi: 10.2147/JPR.S377789. PMID: 36147455; PMCID: PMC9488611.

·         Sipowicz K, Podlecka M, Mokros Ł, Pietras T. Lonely in the City-Sociodemographic Status and Somatic Morbidities as Predictors of Loneliness and Depression among Seniors-Preliminary Results. Int J Environ Res Public Health. 2021 Jul 6;18(14):7213. doi: 10.3390/ijerph18147213. PMID: 34299666; PMCID: PMC8305915.

Additionally, there is an error in the conclusion section: instead of „but not in the BMI 25 group” it should be “but not in the BMI 25 group”.

Round 2

Reviewer 1 Report

I strongly appreciate the work of the authors in improving and clarifying their data. 

Just check English spelling and try to explain how depressive symptoms and obesity could influence chronic pain.

I do not fully agree  with your statement that obesity-related mechanism may work to a greater extent in chronic pain, since your data seem to suggest that they work in different ways in different patient population, and therefore pain pathogenesis could recognise different mechanisms (obese vs non obese for example). This could have clinical implications for pain treatment, since it could been argued that, for example, antidepressant could have a role in treating chronic pain in patients with depressive symptoms but a BMI<25, but could have no implication in obese patients.

Please clarify this issue, I think it would add more insight in your research and open to further research.

Author Response

I strongly appreciate the work of the authors in improving and clarifying their data.

Point 1: Just check English spelling and try to explain how depressive symptoms and obesity could influence chronic pain.

I do not fully agree  with your statement that obesity-related mechanism may work to a greater extent in chronic pain, since your data seem to suggest that they work in different ways in different patient population, and therefore pain pathogenesis could recognise different mechanisms (obese vs non obese for example). This could have clinical implications for pain treatment, since it could been argued that, for example, antidepressant could have a role in treating chronic pain in patients with depressive symptoms but a BMI<25, but could have no implication in obese patients.

Please clarify this issue, I think it would add more insight in your research and open to further research.

Response 1:

We spell-checked the entire manuscript.

We have amended the Discussion section as follows:

“Therefore, the mechanisms of CP development are different for obesity and DS. On the other hand, our results showed that DS is a factor involved in lumbar/knee pain in non-obese subjects, whereas the presence or absence of DS does not make a difference in the prevalence of lumbar/knee pain in obese subjects. Our findings showed that although obesity and DS are associated with CP, the presence of both factors does not additively or synergistically increase their prevalence. It is considered unique to this study since it has not been shown in previous studies.”